# High temperature methane emissions from Large Igneous Provinces as contributors to late Permian mass extinctions

Chengsheng Chen [1,2,3], Shengfei Qin[4], Yunpeng Wang [1,3] ✉, Greg Holland[5], Peter Wynn [2], Wanxu Zhong[5] & Zheng Zhou [2] ✉

Methane ($CH_4$) emissions induced by Large Igneous Provinces have the potential to contribute to global environmental changes that triggered mass extinctions in Earth's history. Here, we explore the source of methane in gas samples from central Sichuan Basin, which is within the Emeishan Large Igneous Province (ELIP). We report evidence of high methane formation temperatures (between $249^{-17/+19}$ and $256^{-20/+22}$ °C) from clumped methane measurements and mantle-derived signatures of noble gases, which verify that oil-cracked $CH_4$ and pyrobitumen are by-products within the reservoirs, associated with hydrothermal activity and enhanced heating by the ELIP. We estimate the volume of oil-cracked $CH_4$ induced by the ELIP and argue that $CH_4$ emissions would have been sufficient to initiate global warming prior to the end of the Permian. We also suggest that similar emissions from oil-cracked $CH_4$ associated with the Siberian Traps Large Igneous Province may also have contributed to the end-Permian mass extinction significantly.

Major mass extinction events during the last 500 Ma of Earth's history coincide with the eruptions of Large Igneous Provinces (LIPs)[1–4]. They have been attributed to a combination of magmatic activities and greenhouse gas release[2,3,5,6]. During the eruptions, massive quantities of greenhouse gases (e.g., $CO_2$ and $CH_4$) were emitted into the atmosphere, leading to rapid global warming, which then contributed to the widespread demise of both aquatic and terrestrial ecosystems[1,6–10]. The Late Permian mass extinctions (LPME), the most severe biosphere crisis in Earth's history, eliminated more than 90% of the Earth's species. Two independent extinction events during the LPME, the Guadalupian-Lopingian extinction (GLE) and the Permian-Triassic extinction (PTE), occurred within a fairly short period (~10 Ma), which, in timing, could be attributed to the eruption of the Emeishan Large Igneous Province (ELIP; ~260 Ma) and Siberian Traps Large Igneous Province (STLIP; ~252 Ma) respectively[5,8,9,11–15]. High-resolution stratigraphy and paleo-biodiversity studies also support the strong correlation between the LPME and the LIPs[7,15,16]. The links among the

global carbon cycle, climate change, and mass extinctions have been recorded in the 5 to 8 per mil (‰) negative shift in stable carbon isotopes of both carbonate and organic carbon ($\delta^{13}C_{carb}$ and $\delta^{13}C_{org}$) through the LPME[7,10,12].

Although an association between global warming and the LPME has been widely accepted, temporal emission mechanisms of greenhouse gases are not entirely clear and remain a topic of discussion. The volcanism associated with the LPME triggered greenhouse gas outbursts and extreme climate changes[2,5–9,12], primarily due to $CO_2$ from magma degassing, thermal metamorphism, and combustion of coal, carbonates, and shales[1,9,11]. In contrast, $CH_4$ released by volcanic intrusion into coal, destabilization of submarine and permafrost clathrates, and enhanced microbial methanogenesis is thought to have had a secondary effect[1,7,11]. However, large quantities of solid bitumen (or termed as pyrobitumen), the by-product of the pyrolysis of oils into $CH_4$[17,18], can be found in areas within the ELIP and STLIP regions[19–21]. This additional source of methane, generated underground by oil

[1]State Key Laboratory of Organic Geochemistry, Guangzhou Institute of Geochemistry, Chinese Academy of Sciences, Guangzhou 510640, China. [2]Lancaster Environmental Centre, Lancaster University, Lancaster LA1 4YQ, UK. [3]CAS Center for Excellence in Deep Earth Science, Guangzhou 510640, China. [4]Research Institute of Petroleum Exploration & Development, PetroChina, Beijing 100083, China. [5]Department of Earth and Environmental Sciences, the University of Manchester, Manchester M13 9PL, UK. ✉e-mail: wangyp@gig.ac.cn; z.zhou4@lancaster.ac.uk

cracking and subsequently released into the atmosphere, may play a much more important role during the LPME interval than previously thought. $CH_4$ is a potent greenhouse gas, and its global warming potential is approximately 28 times relative to $CO_2$ for a 100-year time horizon without considering climate feedback (e.g., stratospheric ozone depletion)[22-24].

Pyrolysis of paleo-oil and related $CH_4$ emissions induced by the volcanic activity of both the ELIP and STLIP have been investigated previously based on petrological recordings and gas-venting pipes[25-27]. It was suggested that carbon gases ($CO_2$ and $CH_4$) and even halocarbons might have been released into the atmosphere through pipes in both Sichuan Basin, China[25] and Tunguska Basin, Siberia[26] during the Late Permian. Nevertheless, previous work was based on indirect evidence in the petrological recordings to infer the possibility of gas emissions. They did not quantify the extensiveness of the pyrolysis process and had insufficient evidence of its strong correlation with the LIPs. In this study, we focus on natural gases by combining clumped isotopes for methane and isotope tracers for noble gases together with basin modeling techniques, to investigate the link between volcanism and $CH_4$ generated by large-scale oil cracking, and evaluate its impact on the LPME.

The Sichuan Basin (~260,000 $km^2$) in the southwestern China is an ideal case study, because (a) it is located in the outer zone of the ELIP in the Upper Yangtze platform of South China continental block (Fig. 1a); (b) abnormal heating events induced by the ELIP have been identified in the basin[8,13,28] and (c) widespread pyrobitumen and natural gas pools have been discovered within the Sinian-Cambrian dolostones in the basin, indicating that massive in-situ paleo-oils have been completely pyrolyzed into $CH_4$ in its geological history[18,21,29]. The ELIP is partly occupied by the Sichuan Basin and its magmas from the Emeishan mantle plume (EMP) have intruded through the Upper Yangtze sedimentary sequences (i.e., Precambrian to Silurian dolostones, marls, and shales) (Fig. 1a). Large basaltic eruptions, crustal melting, hydrothermal activity, and abnormal heating events occurred in the basin and may have initially covered >500,000 $km^2$ from inner to outer zones[1]. A thick succession of marine carbonates overlies the ~800 Ma old basement, where a large-scale paleo-uplift (Leshan-Longnvsi) formed in the center of the basin and became a large petroleum system over the early Paleozoic[29,30] (Fig. 1b). The sedimentary cover of the paleo-uplift includes Sinian-Ordovician marine carbonates, Permian-Triassic carbonate-clastic rocks and Triassic-Quaternary clastic rocks (Fig. 1c). The Anyue gas field, with an area of 22,000 $km^2$, is located on the high point (Moxi-Gaoshiti Bulge) of the paleo-uplift with proven gas reserves of over one trillion cubic meters within the Sinian Dengying ($Z_2dn$) and Cambrian Longwangmiao ($\epsilon_1l$) Formations. The thick black shale of the Qiongzhusi Formation was deposited within the depression on the west of the high point. As the major source rocks (dominated by Type-I kerogen), the shale generated and expelled oils which migrated into the traps within the high points during the oil generation window from Ordovician to Devonian[31-33]. Subsequently, the accumulated oils were pyrolyzed into pyrobitumen and methane gas, which were the most common fluids filling the pore space in the Sinian-Cambrian reservoirs[30-32]. Geochemical evidence from pyrobitumen indicated that the gas pools were formed by in-situ thermal pyrolysis of paleo-oil pools[18,30].

Although previous studies have proposed that the formation of pyrobitumen within the strata may have been affected by occasional hydrothermal activity associated with the ELIP[18,25,30], their evidence is not sufficient to directly link the massive generation of oil-cracked $CH_4$ within the ELIP to the EMP[26,27,31-34]. It remains uncertain whether the ELIP could have acted as a widespread-impact "coking furnace" for promoting the massive generation of oil-cracked $CH_4$ and pyrobitumen underground. Here, we examine 20 natural gas samples (dominantly $CH_4$) collected from the Sinian-Cambrian dolostones in the Anyue gas field in the central Sichuan Basin within the outer zone of

ELIP (Fig. 1b, c). We measure methane-clumped isotopes (9 samples) to obtain methane formation temperatures for investigation of the abnormal geothermal activity and determine noble gas isotopic compositions (11 samples) to understand the mantle influence on ELIP region. Furthermore, we conduct numerical simulation of basin evolution and hydrocarbon generation for the gas field to assist in constraining methane formation temperature and genesis (see Methods). This work quantifies the ELIP-induced $CH_4$ generation within the reservoirs. The total volume of methane released into the atmosphere from the whole basin during the gas formation period is estimated as 1440 Gt.

## Results and discussion
### Gas composition and geochemical characteristics
The natural gas appears as a typical dry gas with a dryness index ($C_1/\Sigma C_{2-5}$) ranging from 583 to 3019. It contains $CH_4$ ranging from 90.11% to 99.87%, $C_{2+}$ ranging from 0.03% to 0.17%, $CO_2$ ranging from undetectable to 8.36%, $H_2S$ ranging from undetectable to 1.51%, $N_2$ ranging from undetectable to -3.23%, and trace amounts of other gases (Supplementary Table 1). Methane shares similar $\delta^{13}C$ values ranging from −32.2‰ to −34.2‰ and similar $\delta D$ values ranging from −134‰ to −147‰, while ethane shares similar $\delta^{13}C$ values ranging from −27.5‰ to −33.3‰ (Supplementary Table 1). Low $H_2S$ and $CO_2$ content is likely related to minor thermochemical sulfate reduction (TSR) that took place in the reservoirs[18,32]. Almost all gases from the Sinian and Cambrian reservoirs were generated from oil precursors accompanied with abundant pyrobitumen from 0.1 to over 8.0%wt (average approximately 1.0%wt)[17,21]. In addition to extremely high dryness index (≫10), $\delta^{13}C$ and $\delta D$ values (much higher than −60‰ and −300‰, respectively) also show that gases from the Sinian and Cambrian reservoirs have reached thermal equilibrium. They fall within the equilibrium thermogenic field[35] defined by using $\delta^{13}C$ and $\delta D$ values (Supplementary Fig. 6a), indicating that there are no influences from biogenic or abiotic gases[35-37]. Extremely high dryness index, a single in-situ gas source, and equilibrium conditions permit the use of the $\Delta^{13}CH_3D$ thermometer (defined by equation 4 in the Supplementary Information) for deriving formation temperatures of methane.

### Constraining methane formation temperatures in the Anyue field using $\Delta^{13}CH_3D$ values
$\Delta^{13}CH_3D$ values indicate methane formation temperatures in the $Z_2dn$ samples vary from $248^{-16/+17}$ °C to $269^{-20/+22}$ °C (average of $256^{-20/+22}$ °C), while the $\Delta^{13}CH_3D$-based temperature values of methane in $\epsilon_1l$ samples vary from $246^{-15/+17}$ to $250^{-17/+19}$ (average of $249^{-17/+19}$ °C) (Fig. 2b; Supplementary Table 2). These temperatures are significantly higher than the present reservoir temperatures (140–165 °C)[21,38] and the peak temperatures of either TSR (~140 °C)[34,39] or oil-cracking (160–180 °C)[40,41] (Fig. 2b; Supplementary Fig. 3). Also, they are significantly higher than the modeled reservoir temperatures (200–220 °C) of the maximum burial during Late Cretaceous (Fig. 2a, b). In contrast, clumped methane isotope temperatures are closer to the highest trapping temperatures (249–319 °C) of the quartz inclusions that have been reported in the same gas reservoirs in the Anyue gas field[30], representing the invasion of deep-to-epizonogenic hydrothermal fluids corresponding to the ELIP, EMP[18,25] (Fig. 2b). These generally higher temperatures derived from $\Delta^{13}CH_3D$ values indicate significant hydrothermal control in addition to the thermal effect associated with burial process.

The interpretation of high-temperature methane formation in the Anyue gas field is supported by the gas compositions. Studies have shown that temperature differences can generate completely different end products especially in organic reactions[42]. The kinetics of the oil-cracking process has two distinct stages with significant difference in gas composition of methane ($C_1$) and heavy hydrocarbon gases ($C_{2-5}$)[40,41]. The first stage is characterized by dominant production of

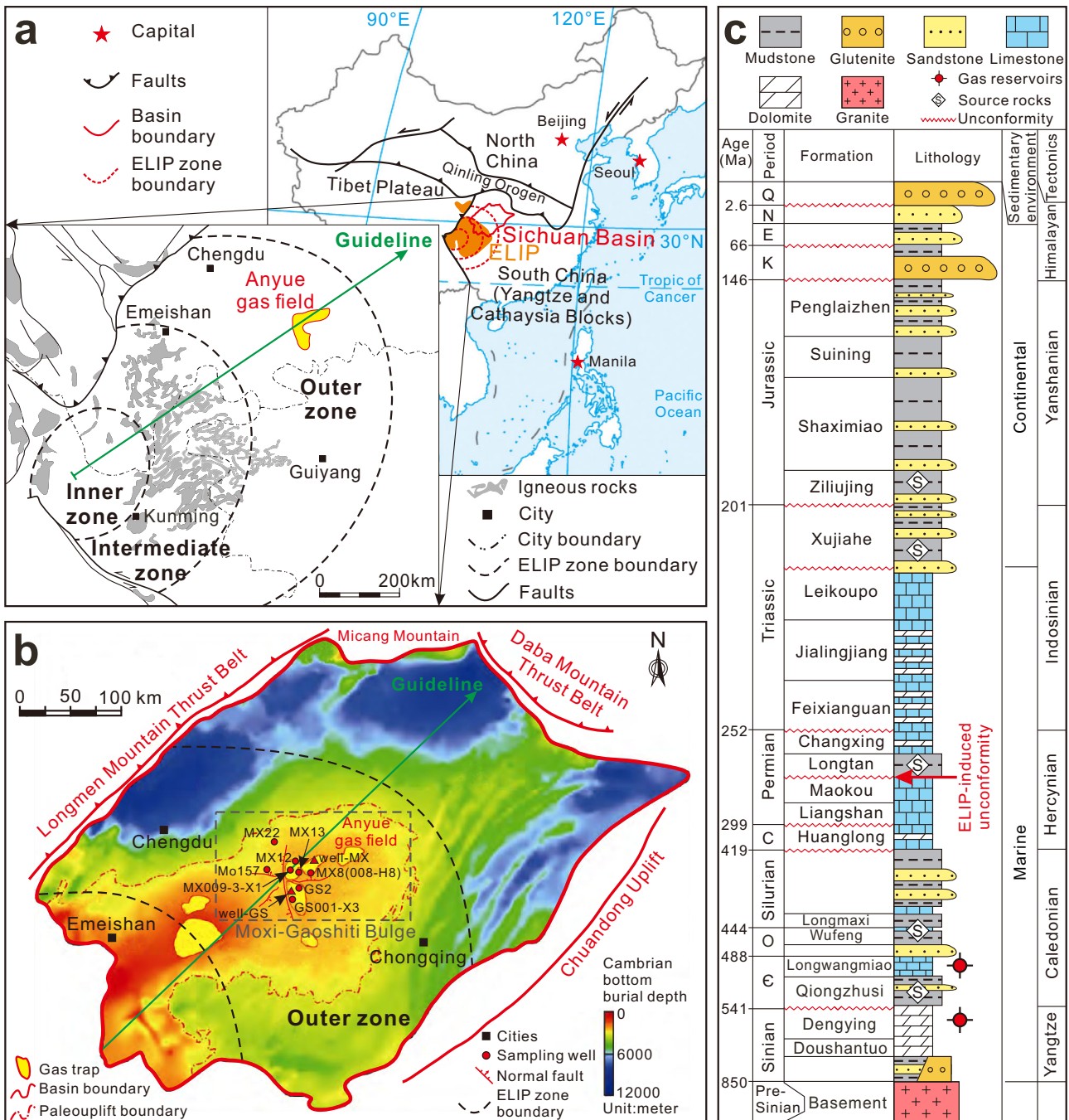

**Fig. 1 | Tectonic and stratigraphic background of the study area and adjacent regions. a** Map showing the geographic distribution of the Emeishan Large Igneous Province (ELIP) and Sichuan Basin in the Upper Yangtze platform, southwestern China[13,28]. The administration boundaries in the map are originated from map products of National Geomatics Center of China (http://www.webmap.cn). **b** Tectonic setting of the study area and adjacent regions. The map of burial depth depicts the Cambrian bottom and constraints the boundary of Leshan-Longnvsi paleo-uplift[29]. Nine gas samples for methane clumped isotope and noble gas analysis are indicated, samples not shown in the map are also collected from the same area in Sinian-Cambrian reservoirs. They are located on the Moxi-Gaoshiti Bulge as part of the Leshan-Longnvsi paleo-uplift, which is the center of the basin as well as the outer zone of ELIP. **c** Generalized stratigraphic column of the Sichuan Basin[65].

Marine environments persisted in the basement from the Sinian to the Middle Triassic controlled by the Yangtze, Caledonian, and Hercynian movements, occurring unconformities of the Sinian-Cambrian by the deformation, the Devonian-Carboniferous by the late Paleozoic lifting, and the Permian-Triassic by the ELIP-induced dome and lifting[13,28,65]. After the tectonic evolution during the middle Triassic, the terrestrial succession had become the main sedimentary facies until Late Cretaceous controlled by the Indosinian and Yanshannian movements[33,66]. The Yanshannian-Himalayan lifting led to extensive absence of the Cenozoic in the basin[32,66]. Symbols used in the figure include Є – Cambrian; O – Ordovician; C – Carboniferous; K – Cretaceous; E – Paleogene; N – Neogene; Q – Quaternary.

$C_{2-5}$ wet gases and pyrobitumen, whereas the second stage is characterized by re-cracking of the $C_{2-5}$ wet gases to methane. This process leads to a progressive increasing dryness index of the gas[17,40]. Kinetic modeling of the $C_{2-5}$ gases showed that the maximum yield of the $C_{2-5}$ gases was from 168 °C to 185 °C at geological heating rates from 2 °C/Ma to 10 °C/Ma and they were completely pyrolyzed into methane at a temperature of ~265 °C[40,42]. Our methane generation model (Fig. 2 and also see Methods and Supplementary Fig. 3) which is based on the

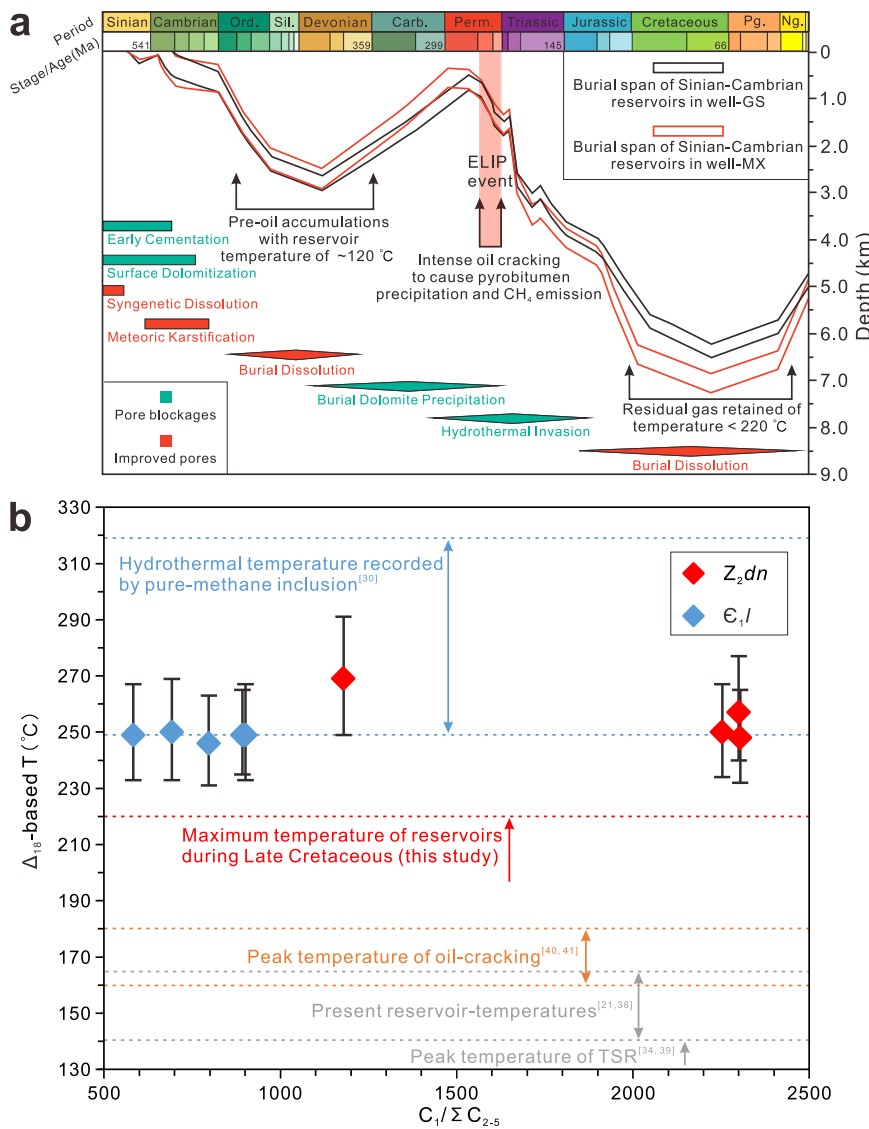

**Fig. 2 | History of geological evolution matches modeled geological temperatures. a** The burial history and modeled reservoir temperatures ($\epsilon_1 l$ and $Z_2 dn$) in the Anyue gas field, central Sichuan Basin recovered by model simulations on well-MX and well-GS (see Supplementary Fig. 1 and Supplementary Fig. 2). The reservoirs with dolomite diagenesis have allowed storage of the pre-existing oils and subsequent oil-cracked gases[30]. The modeled temperatures correspond to the geological evolution history in the successive order of early cementation, surface dolomitization, syngenetic dissolution, meteoric karstification, burial dissolution, burial dolomite precipitation, hydrothermal invasion, and deepest burial dissolution[30,66]. **b** Diagram of $\Delta^{13}CH_3D$-based temperatures ($\Delta_{18}$-based T) vs. $C_1/\Sigma C_{2-5}$ ratios. Dashed lines specify possible formation temperatures at which methane can be generated under different conditions[21,30,34,38–41]. The error bars for $\Delta^{13}CH_3D$-based temperatures are dominantly derived from ±0.1‰ standard deviation for a constant offset against the stochastic distribution (see Supplementary Information).

kinetic model and a geological heating rate of 5.0 °C/Ma for the Sichuan basin indicates that such extremely dry gases ($C_1/\Sigma C_{2-5} = 583$–3019) in the Sinian-Cambrian reservoirs would require a formation temperature beyond 250 °C. In contrast, the formation temperature at the maximum burial depth during Late Cretaceous had not exceeded 220 °C (Fig. 2a, b). At this temperature, it was unlikely for the oils in the reservoirs to form such extremely dry gases. The gas products would have been characterized by high content of wet gases, which contradicts the gas compositions observed. Therefore, evidence from gas composition, kinetic modeling and the $\Delta^{13}CH_3D$-based temperatures support the impact of the ELIP on the formation of methane in the Anyue gas field, by triggering abnormal heating and rapid oil cracking.

High-temperature methane formation in the Anyue gas field can also be supported by petrological evidence. Optical characteristics of pyrobitumen in the $\epsilon_1 l$ and $Z_2 dn$ reservoirs[18,25,30] were observed to be similar to mesophase pitch, a liquid crystal material produced at high

temperatures in a coking furnace, indicating that organic matter had been transferred to a graphite crystal. Honeycomb micropores, generally observed in carbon foams, appeared in the pyrobitumen[18], suggesting that the reservoirs had undergone a coking process by hydrothermal fluid invasion with rapid heating rather than gradual burial in the geothermal history. These anomalous temperatures suggest that methane generated in the Anyue gas field was mainly controlled by the invasion of hydrothermal fluids[18,30] during the ELIP interval rather than the maximum burial at the Late Cretaceous, as such high temperatures in the region are only available during the ELIP period (Fig. 2). Further investigation on the mantle involvement is carried out by using noble gas tracers.

## Tracing ELIP-related hydrothermal activity in the Anyue gas field using noble gases

Mantle-derived hydrothermal fluids would be expected to retain a mantle signature in He isotopes (Supplementary Table 3 and

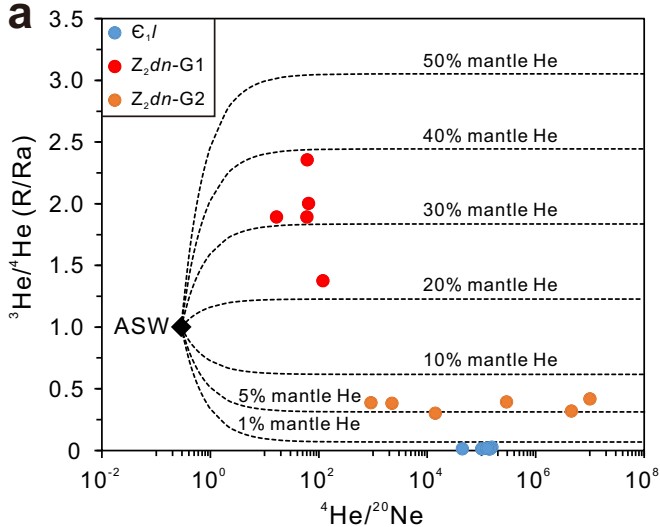

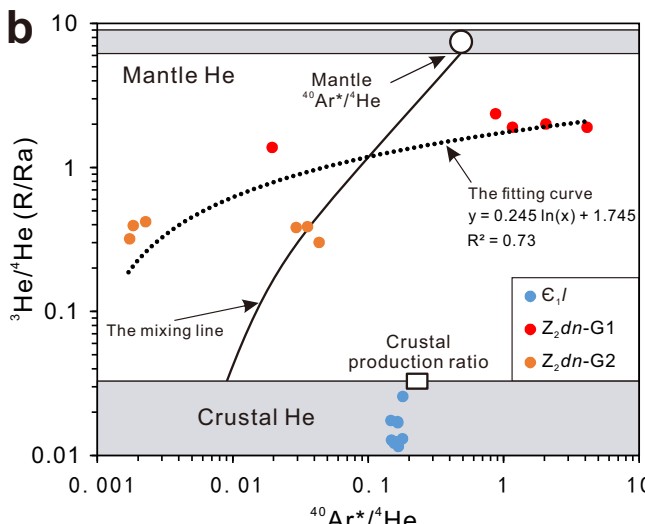

**Fig. 3 | Noble gas signatures support the link of methane with the mantle influence during the Emeishan igneous period. a** Diagram of $^4$He/$^{20}$Ne vs. $^3$He/$^4$He ($R/R_a$) ratios. The $^3$He/$^4$He ratio $R$ is normalized to the atmospheric $^3$He/$^4$He ratio ($R_a = 1.384 \times 10^{-6}$)[67]. Measured $^4$He/$^{20}$Ne ratios in all samples ranged from 16.5 to $9.96 \times 10^6$. They are much higher than the ratio in air-saturated water (ASW = 0.288)[68] or air (0.318)[67]. Therefore, the atmospheric or ASW-derived gas has negligible contribution to the He concentrations. To estimate mantle He contributions, a simple two-component mixing model[44] was used between an upper crustal endmember (0.008$R_a$)[48] and a mantle endmember relative to the sub-continental lithospheric mantle (SCLM) (6.1 ± 0.9$R_a$)[69]. **b** Diagram of $^{40}$Ar*/$^4$He vs.

$^3$He/$^4$He ($R/R_a$) ratios. $^{40}$Ar* represents the resolved non-atmosphere derived excess $^{40}$Ar. The extrapolated mixing line between the crust and mantle endmembers was defined by Stuart et al. using unfractionated cases measured in Dae Hwa (South Korea) W-Mo deposit fluid inclusions[44,49]. The mantle $^{40}$Ar*/$^4$He = 0.69 ± 0.06 is typical of unfractionated samples from the mantle[50]. In contrast, the crustal $^{40}$Ar*/$^4$He = 0.007 is far lower than the crustal production ratio of ~0.2, typically representing a fluid derived from shallow cool regions of the crust[48], in which crustal fluids often mix with a He-rich endmember due to preferential addition of $^4$He to the gas phase[44].

Supplementary Table 4). Measured $^3$He/$^4$He ratios ($R/R_a$, where $R_a$ is the atmospheric value of $^3$He/$^4$He) range from 0.0115 to 0.0256 in $\in_1 l$ samples showing crustal dominance of helium isotopes[43], while the $R/R_a$ values in samples from Z$_2$dn-G1 and Z$_2$dn-G2 range from 1.37 to 2.36 and 0.300 to 0.418, respectively, suggesting a strong mantle signature[44]. Here G1 and G2 are two subgroups of samples from the Z$_2$dn reservoir, taking 1$R_a$ as the separation standard. Because the ELIP induced by the EMP was a mafic continental large igneous province developed in a typical non-rifted continental margin[13,28], we can use He ratios in the subcontinental lithospheric mantle (SCLM) as the mantle endmember of He for further study. Therefore, the mantle contribution of helium can be resolved. The results show that He in $\in_1 l$ samples is almost entirely crustal derived (>99.7%), with the mantle contribution <0.30%. However, He in Z$_2$dn-G1 and Z$_2$dn-G2 samples are a mixture of mantle and crustal He in various proportions: The mantle $^3$He contributions in Z$_2$dn-G2 samples vary from 4.80% to 6.74%. For Z$_2$dn-G1 samples, the mantle $^3$He contributions are ranging from 22.43% to 38.54% (Fig. 3a). He signatures in all samples suggest that, although long-term cratonic stability of the Sichuan basin basement prevented significant volatile contributions from the mantle[45,46], isolated $^3$He "hotspots" do exist and are likely associated with the ELIP[18,30].

Measured $^{36}$Ar concentrations among all samples are diverse and $^{40}$Ar/$^{36}$Ar ratios show clear deviation from the atmospheric $^{40}$Ar/$^{36}$Ar ratio of 298.6[47]. Measured $^{40}$Ar/$^{36}$Ar ratios vary from 2168 to 5973 in $\in_1 l$ samples, from 288 to 348 in Z$_2$dn-G1 samples, and from 323 to 13558 in Z$_2$dn-G2 samples. Resolved non-atmosphere derived excess $^{40}$Ar ($^{40}$Ar*) contributes from 86.2% to 95.0% of the measured $^{40}$Ar concentrations in $\in_1 l$ samples, from 0.3% to 14.7% in Z$_2$dn-G1 samples, and from 7.5% to 97.8% in Z$_2$dn-G2 samples. All $^{40}$Ar*/$^4$He values in $\in_1 l$ samples range from 0.148 to 0.181, which is much higher than those found in a fluid derived from shallow cool regions of the crust, where $^{40}$Ar*/$^4$He = 0.007, but close to the crustal production ratio of 0.2[48] (Fig. 3b), implying the contribution from hotter (or deeper) regions of the crust where the diffusivities of noble gases are sufficiently high to permit the

release of both He and Ar. From an extrapolated mixing line of $^{40}$Ar*/$^4$He vs $^3$He/$^4$He ($R/R_a$) between the crust and mantle endmembers[44,49], the $^{40}$Ar*/$^4$He values of Z$_2$dn-G1 (from 0.020 to 4.140) and Z$_2$dn-G2 (from 0.002 to 0.044) samples indicate the mixing of the crustal and mantle components, implying diverse $^{40}$Ar contributions from the mantle or possible fractionation between He and Ar by gas, water, and rock interactions (Fig. 3b). A natural logarithmic fitting curve for $^3$He/$^4$He and $^{40}$Ar*/$^4$He ratios in the Z$_2$dn samples ($R^2 = 0.73$) shows that higher $^3$He/$^4$He ratios correlate with higher $^{40}$Ar*/$^4$He values. However, this curve is different from the extrapolated mixing line. Sample $^{40}$Ar*/$^4$He values are either lower than the value for the shallow cool regions of the crust ($^{40}$Ar*/$^4$He = 0.007)[48] or beyond the mantle production ratio ($^{40}$Ar*/$^4$He = 0.69 ± 0.06)[50] (Fig. 3b).

The Cambrian reservoirs ($\in_1 l$) contain significant crustal noble gas components. This may be related to its much thicker underlying depositional systems (carbonates and shales), resulting in higher lateral fluxes of crustal radiogenic He, but not permitting efficient vertical flux of mantle-derived gases (e.g., $^3$He) to the present gas reservoir (Fig. 4d, e). Compared to the Cambrian reservoir, the Sinian reservoirs (Z$_2$dn) which recorded significant mantle-derived He, directly overlie the basement and have thinner underlying depositional formations which facilitate easier transit of mantle-derived gas and lower contributions of He from crust. However, samples with intermediate $^3$He/$^4$He ratios in the Z$_2$dn-G2 have highest $^4$He concentrations. This demonstrates variable contributions of both mantle and crustal He in the Sinian formations close to the basement, likely indicating a heterogeneous upwelling of mantle fluids associated with a deeper thermal pulse. Specifically, the Sinian reservoirs (Z$_2$dn) exhibit high $^{40}$Ar*/$^4$He values in the Z$_2$dn-G1 samples and low $^{40}$Ar*/$^4$He values in the Z$_2$dn-G2 samples (Fig. 3b). The closure temperature for $^{40}$Ar* in minerals is much higher than that of He[48,50,51]. Therefore, He may be preferentially released from minerals at low temperature in the Z$_2$dn-G2 reservoir and Ar may be preferentially released from minerals at high temperature in the Z$_2$dn-G1 reservoir. High $^{40}$Ar*/$^4$He ratios suggest that

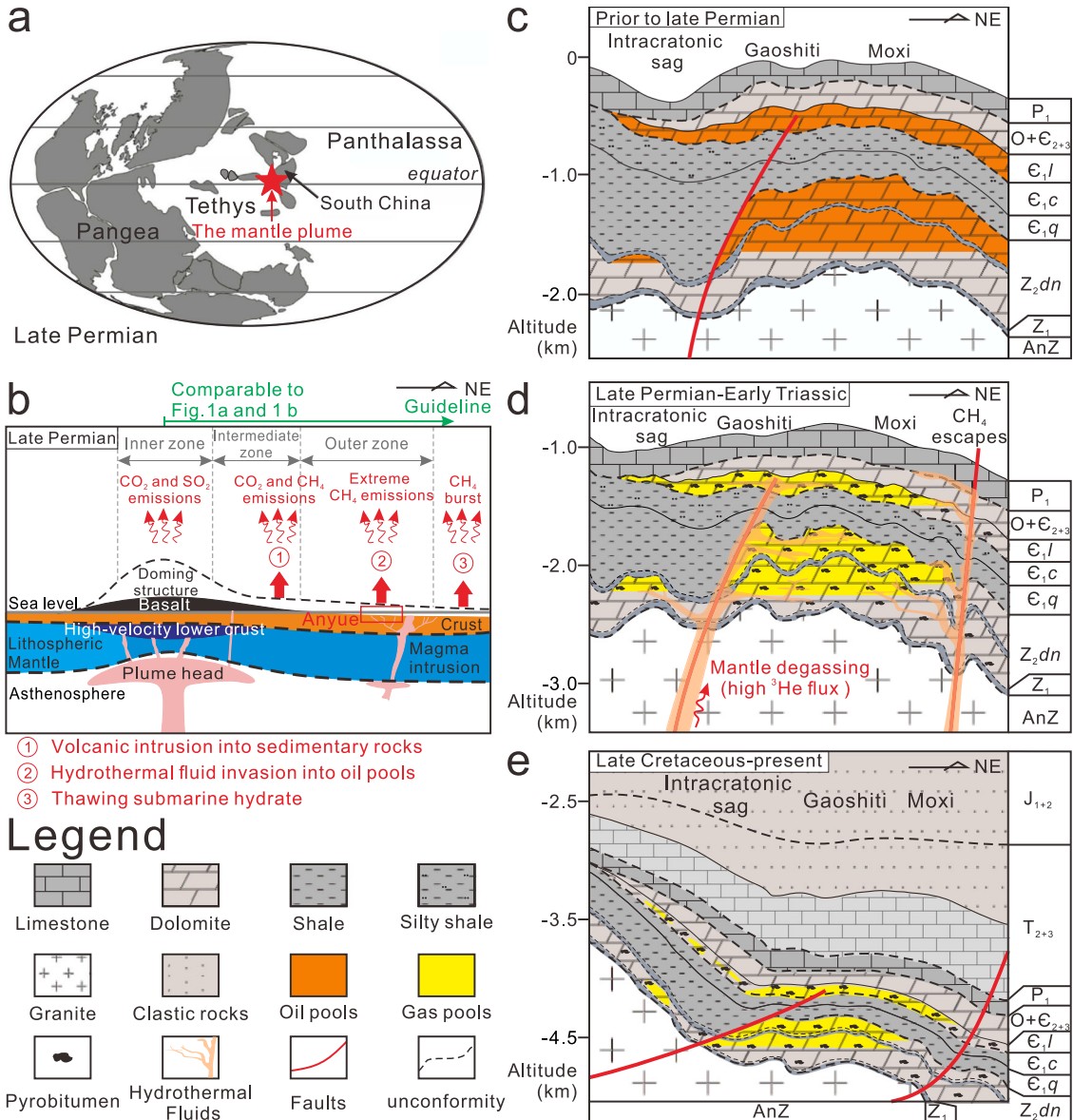

**Fig. 4 | The pattern of methane production and emission in the central Sichuan Basin, China. a** Global paleogeographic map during the Late Permian-Early Triassic period[16,70] and the locations of Sichuan Basin and mantle plume. **b** Surface uplift, generation of Emeishan basalts, crustal accretion in context of upwell of mantle plume[28], and greenhouse gas emission patterns. **c** The formation of paleo-oil pools prior to late Permian[29]. The Qiongzhusi Formation ($\mathcal{C}_1 q$) mainly distributed at the intra-cratonic sag is the main source rock to provide hydrocarbon sources[18,29,38]. **d** The hydrothermal fluids associated with the magma events invaded the oil reservoirs during the Late Permian-Early Triassic through hydrothermal channels caused by pull-apart function of strike-slip faults[53], which caused the oil-cracking, precipitation of hydrothermal minerals[18,30], methane emissions to ancient

atmosphere, and mantle-degassing. **e** The basin experienced continuous deposition until late Cretaceous. Then the extensive tectonic uplift occurred due to the Himalayan orogeny. The differential extent of uplift created a geological pattern with west of the basin higher than the east of basin. This geological adjustment further isolated the reservoirs resulting in good traps. It provided unique conditions for long-term retention of natural gases and mantle-derived gases especially in the $Z_2 dn$ reservoirs. Symbols used in the figure include AnZ – Pre-Sinian; $Z_1$ – Doushantuo Formation; $Z_2 dn$ – Dengying Formation; $\mathcal{C}_1 q$ – Qiongzhusi Formation; $\mathcal{C}_1 c$ – Canglangpu Formation; $\mathcal{C}_1 l$ – Longwangmiao Formation; $\mathcal{C}_{2+3}$ – Middle-Upper Cambrian; O – Ordovician; $P_1$ – Lower Permian; $T_{2+3}$ – Middle-Upper Triassic; $J_{1+2}$ – Lower-Middle Jurassic.

the $Z_2 dn$-G1 reservoir has received high ³He and ⁴⁰Ar* fluxes close to fractures or vents, and experienced more intense hydrothermal activity, leading to higher temperatures of surrounding rocks, which facilitated the release of ⁴⁰Ar* from minerals. Variation of ⁴⁰Ar*/⁴He ratios also suggests the presence of abnormal thermal events and the heterogeneity of associated hydrothermal activity. Generally, fluid inclusions in minerals (e.g., carbonates) cannot trap gases at temperatures greater than 250 °C[51]. At high temperatures, noble gases would diffuse quantitatively into free gas phases producing high ⁴⁰Ar*/⁴He ratios[48]. Based on the ⁴⁰Ar*/⁴He ratios in Cambrian reservoirs (Fig. 3b), the maximum temperature in this isolated formation would

be higher than 250 °C, which is consistent with the clumped isotope temperatures.

## The extent of methane production and emission induced by ELIP

Figure 4 shows the pattern of methane production and emission in the central Sichuan Basin induced by ELIP during its geological evolution. It suggests close association with mass extinction caused by global warming. Prior to late Permian (Fig. 4c), the high-quality source rocks of Qiongzhusi Formation ($\mathcal{C}_1 q$) developed in the intra-cratonic sag entered the oil generation window of formation temperatures over

120 °C during the Ordovician-Devonian period[31–33]. This supplied sufficient hydrocarbons which then formed large oil pools within high-quality $\in_1 l$ and $Z_2 dn$ reservoirs at the high points of the Moxi-Gaoshiti Bulge, central Sichuan Basin. Due to stable geological conditions and good seals, these oil pools were well-preserved until the late Permian ELIP. During late Permian (Fig. 4d), corresponding to the period of ELIP activity (Fig. 4a–c), high-temperature hydrothermal fluids (~319 °C)[30] invaded the basin through faults, unconformities, and high-porosity reservoirs, and then efficiently pyrolyzed the oils into methane and pyrobitumen within the reservoirs[31,32,52]. This process has been recorded in in-situ pyrobitumen, hydrothermal veins, quartz inclusions and other hydrothermal-related minerals[18,21,30,33]. Because the abnormal thermal event caused by the ELIP ($259.1 \pm 0.5$ Ma)[14] might have peaked within 1–5 Ma[4], the timing of oil pyrolysis coincided with the GLE (~260 Ma). The abnormally high clumped temperatures strongly support this scenario, and high $^3He/^4He$ isotopic ratios record the link between the basin and mantle during the period of ELIP.

We propose total methane generation (TMG) by oil cracking based on the total content of pyrobitumen and its yield (see Methods and Supplementary Table 5) is up to $2.01 \times 10^{14}$ m$^3$ in the Anyue region. Excluding preserved methane (total gas reserves, TGR), the total methane emission (TME) into the paleo-atmosphere could have been $2.00 \times 10^{14}$ m$^3$ or 144 Gt (>99% of total gas generated) (see Methods and Supplementary Table 5). Such a large amount of methane could be emitted through hydrothermal channels and fracture systems triggered by late Permian volcanic and seismic activities, and its release could be driven by rapid gas expansion and escape (Fig. 4d). Such channels and faults have been revealed in previous research based on seismic data. For example, the Sinian Dengying Formation contains widely developed hydrothermal channels that formed on the basis of tiny-grabens caused by the pull-apart function of strike-slip faults[53], which likely provided potential faults and pipes contemporaneous with the ELIP for gas venting[25]. Such rapid expansion and extreme high pressures have been preserved in CH$_4$-dominated gaseous inclusions within the quartz formed in the reservoirs[52]. Paleo-pressure coefficients of larger than 3.0 were recorded under Late Permian trapping depths compared to the values of the gas pool pressure coefficients of less than 1.2 under current reservoir conditions[21,30]. This indicates that the Anyue gas field has experienced rapid oil cracking, gas expansion, and escape processes induced by the ELIP. Gas generation by burial only played a minor role in the Anyue region.

The total volume of pyrobitumen in the Sichuan Basin and its surrounding area is over ten times that of the Anyue region[17,18]. Therefore, the entire Sichuan Basin could have released 1440 Gt CH$_4$, which is equivalent to at least 40,410 Gt CO$_2$, as the global warming potential of CH$_4$ is at least 28 times that of CO$_2$ in a 100-year period[22,24]. This is comparable to previous work which linked the GLE event with 16,800 Gt CO$_2$ (11,200 Gt from magma release and 5600 Gt from magma intrusions) emitted by the ELIP volcanism[1]. The mass of oil-cracked CH$_4$ equivalent to 40,410 Gt CO$_2$ suggested in this work is more than twice as large as the CO$_2$ released by the ELIP itself, or over one thousand times the annual global carbon emission by human beings (e.g., 36 Gt CO$_2$ emission in 2019)[54]. This indicates that CH$_4$ induced by the ELIP could be a key driver of global warming and climate change during the GLE event, without taking climate feedbacks into account, such as stratospheric ozone depletion by methane chemical loss[22,23].

### Significance of high-temperature methane emissions to the paleoclimate, LPME, and global carbon cycle

The PTE event is the largest known extinction event, coincident with the greatest STLIP volcanism, which happened 8 million years after the ELIP[1,7]. The STLIP volcanism may have had a magma volume 6–10 times of the ELIP[55,56] in a region that had accumulated large-scale Pre-cambrian-Paleozoic sediments containing abundant pyrobitumen[19,20],

in a geological setting and sedimentary framework similar to the Upper Yangtze plate (e.g., Sichuan Basin). Therefore, significantly more CH$_4$ (probably over $10^4$ Gt) could have been generated by the STLIP, implying CH$_4$ emissions may directly have contributed to global warming and the greatest mass extinction event at the end of the Permian.

Furthermore, more than one-third of the erupted volcanic rocks and the entire STLIP intrusive magmatism postdated the end-Permian mass extinction (EPME)[2,7,57], reducing the likelihood that CO$_2$ release is solely responsible for the mass extinction. Although elemental and isotopic signatures of Cu and Hg have verified that felsic volcanism (S-rich vapor release) in South China might have accelerated the process of extinction[7], the oil-cracked source of CH$_4$ proposed in this study should also be recognized as an important driver. LIPs can cause abnormal heating events and hydrothermal fluids before, during, and after their massive eruptions[8,13,28]. The relatively low temperature (<300 °C, see Supplementary Fig. 3) required for rapid oil cracking to form pyrobitumen and CH$_4$, can be achieved before CO$_2$ emission from massive basalt eruptions and magma intrusions with temperatures of ~400–1200 °C[1]. This implies that CH$_4$ generation and emission could occur before CO$_2$, and more likely be responsible for the EPME event, at least in the early stages. Similarly, the massive emission of CH$_4$ (about 7200 Gt) that has been estimated to occur during the end-Triassic Central Atlantic Magmatic Province (CAMP)[58], likely also contributed to global climate change which caused the end-Triassic mass extinction[58,59], which is comparable to ELIP's 1440 Gt and SLITP's $10^4$ Gt CH$_4$ emissions. This further supports the concept that high-temperature CH$_4$ emissions induced by LIPs are of major importance for understanding mass extinctions and the carbon cycle in Earth's history.

In this study, we propose that high-temperature methane generated by magmatic activities is an important driver for climate change and carbon cycle. It is different from thermogenic methane produced from organic matter by geological burial processes. It is also different from abiotic methane associated with Fischer-Tropsch reactions[60]. This type of high-temperature methane is still thermogenic in origin, but it is produced by geothermal anomalies related to magmatic and hydrothermal activity rather than sedimentary processes. The unique feature of high-temperature methane is the extremely rapid cracking of organic matter caused by rapid and extreme heating events which contrasts with slow heating and therefore slow methane generation during sedimentary burial. Although the samples in this study are collected from the outer zone of the ELIP, they still show significant impact from the heat related to the ELIP. Samples closer to the intermediate and even inner zone (Fig. 1a; Fig. 4b), would exhibit even more rapid oil pyrolysis and high-temperature methane release would be greater still.

On a global scale, oil-cracked CH$_4$ emissions likely have contributed significantly to elevated CH$_4$ concentrations in the atmosphere in Earth's history as an important part of the global carbon cycle. In addition to the effect of global warming, methane, as a reactive gas, is a precursor to other pollutants in the atmosphere (e.g., CH$_3$Cl), which have significant impact on the biosphere (e.g., ozone depletion)[22,23]. Therefore, high-temperature methane emissions require much more attention. In Earth's history, large amounts of high-quality source rocks and thick carbonate rocks formed large numbers of paleo-oil reservoirs before the Permian[17–21,30–33]. The volcanic activity can then induce the rapid destruction and secondary cracking of the paleo-oil reservoirs, leading to methane releases that contributed to climate change and mass extinction events, particularly in the Late Permian. This potential source of methane is critical for our understanding of the evolution of the Earth's history. Further understanding can be gained by focusing on methane emissions associated with LIP-induced oil cracking and investigating the global distribution and abundance of reservoir pyrobitumen in the relevant areas of global LIPs.

## Methods

### Samples and laboratory analysis

Natural gas samples were obtained from $Z_2dn$ and $\in_1l$ reservoirs via wellheads distributed in the Anyue gas field (Fig.1b, c). Samples for noble gas and $CH_4$ clumped isotope analysis were collected using 10 mm diameter internally polished refrigeration grade copper tubes sealed with stainless steel pinch-off clamps on both ends[61]. Additional samples for the analysis of gas composition, carbon and hydrogen isotopes were collected using stainless steel cylinders[61]. Details of all samples are available in the Supplementary Information.

Bulk gas and stable isotope analysis were undertaken at the State Key Laboratory of Organic Geochemistry (SKLOG), Guangzhou Institute of Geochemistry, Chinese Academy of Sciences (GIGCAS), using previously described techniques[62,63]. Bulk gas content as a percentage was determined using an Agilent 6890 N gas chromatograph-mass spectrometer (GC-MS)[63]. Carbon and hydrogen isotope values were measured following established procedures by using GC-IRMS of Agilent 6890N-Isoprime 100 and Thermofisher 1310-Delta V, respectively[62,63]. Precision and reproducibility are typically better than ±0.2‰ for $\delta^{13}C$ (PDB) and ±2‰ for $\delta D$ (SMOW). Data used in this study are reported in Supplementary Table 1 and Supplementary Table 6.

Methane-clumped isotope and noble gas analysis were undertaken at the Subsurface Fluid Isotope Geochemistry Laboratory (SFIGL) at Lancaster University, UK. Methane clumped isotopologues ($^{13}CH_4$, $^{12}CH_3D$, $^{13}CH_3D$, and $^{12}CH_4$) were measured by a tunable infrared laser direct absorption spectroscopy (TILDAS, Aerodyne) that houses two continuous wave quantum cascade lasers and directly connects a methane purification system (Inlet system, Protium MS, UK) (Supplementary Fig. 4). The experimental procedure is described in Supplementary Fig. 4. Spectral acquisition and data processing were carried out by using TDL Wintel software[64]. All measurements were made in reference to the lab's working reference gas (LEC-1) taken from a high purity (>99.99%) methane gas cylinder after conducting systematic calibration (see Supplementary Information). Preparation for standard calibration gas and its gas characteristics are described in Supplementary Information (Supplementary Fig. 5, Supplementary Fig. 6, and Supplementary Table 6). All measurements reported in this paper were obtained at a nominal cell pressure of ca. 1.0 Torr. Data of reservoir samples used in this study are reported in Supplementary Table 2.

Noble gas abundance and isotopic ratios were determined by an Isotopx NGX noble gas mass spectrometer at the SFIGL using previously described techniques[43]. The $^4He$, $^{20}Ne$, and $^{40}Ar$ isotopes were measured using a Faraday detector while the remaining isotopes were counted on an electron multiplier. The noble gas elemental abundances for each sample were calculated by normalizing to those of air standards after blank correction. Due to low abundance of He in the atmosphere, the air standard for noble gas measurements at the lab was taken from a gas cylinder filled with a mixture of He spike and Air. The He spike is HESJ-standard with the estimated air normalizing $^3He/^4He$ ratio of 20.63 ± 0.10[43]. During Ne abundance and isotope analysis, appropriate mass peaks were monitored to correct for interferences caused by doubly charged ions of $^{40}Ar^{++}$ and $^{44}CO_2$ on $^{20}Ne$ and $^{22}Ne$, respectively[43]. $^4He$, $^{20}Ne$, and $^{36}Ar$ abundances had typical uncertainties of 1.6%, 1.7%, and 1.7%, respectively. Data used in this study are reported in Supplementary Tables 3 and 4.

### Numerical simulation

Models for well-GS of Gaoshiti bulge and well-MX of Moxi bulge (Fig. 1b) were developed by using the software PetroMod version 2016.2, Schlumberger Company, which fundamentally recovered the burial-thermal history of Anyue gas field, central Sichuan Basin. In each well, the primary input parameters for building the model include the stratigraphy (lithology, thickness, and age), tectonic events (unconformities, erosion and age) and boundary conditions (paleo-water depth, history of heat flow and sediment-water interface temperature). Measured temperature and vitrinite reflectance values were used to validate the modeling results (Supplementary Fig. 1b; Supplementary Fig. 2b). The geological model of each well was completed once the modeled and measured results were consistent. Both burial-thermal diagrams are reported in the Supplementary Information (Supplementary Fig. 1a; Supplementary Fig. 2a) and target output is included in Fig. 2a, b and Supplementary Fig. 3a.

Generation models were calculated based on a kinetic model of Tang (2011) _ SARA_TI that comes with the software PetroMod version 2016.2 and can represent type-I kerogen (sapropel type organic matter) which dominated the Qiongzhusi ($\in_1q$) source rocks[31–33]. The geological heating rate of 5.0 °C/Ma was taken from the maximum of the central basin (Supplementary Fig. 3a). The model results are reported in Supplementary Fig. 3b.

### Methane emission model

Total methane emissions (TME) during the interval of LIPs are equal to total methane generations (TMG) minus total gas reserves (TGR). The TMG value can be estimated on the basis of methane and pyrobitumen yields of oil-cracking, because pyrobitumen can represent a potential bridge between gases and hydrocarbon precursors (e.g., oils)[17]. Previous work has investigated the relationship between pyrobitumen and methane yields by using artificial pyrolysis experiments of crude oils, showing a near-linear correlation with a slope of 1.09 between the yields from high to overhigh maturity (i.e., Easy%$R_o$ > 1.5%)[17]. This slope is equal to the conversion ratio ($R$) between the volume of oil-cracked methane (in L) and the amount of oil-cracked pyrobitumen (in g), which is $R = 1.09$ L/g. Then, the gas generation intensity (GGI, i.e., amount of gas generation per unit area)[17] can be calculated for assessing the scale of oil-cracked methane.

The GGI values can be determined as follows:

$$GGI = H \times C \times D \times R \times 10^7 \text{ m}^3/\text{km}^2 \quad (1)$$

Where $H$ is reservoir thickness (m), $C$ is pyrobitumen content (wt%), $D$ is rock density (g/cm$^3$), $R$ is conversion ratio (i.e., the yield ratio of methane and pyrobitumen, L/g), and the coefficient $10^7$ is for unit conversion.

The TMG value can be calculated by using the GGI value and the pyrobitumen distribution area ($A$) as follows:

$$TMG\,(m^3) = GGI\,(m^3/km^2) \times A\,(km^2) \quad (2)$$

Eventually, the TME values are estimated as follows:

$$TME\,(m^3) = TMG\,(m^3) - TGR\,(m^3) \quad (3)$$

Where the TGR value can be referred to in available results from geological survey. All data used in this study and results of calculations are listed and introduced in Supplementary Table 5.

## Data availability

The data that support the findings of this study are available within the article and the Supplementary Information file. All data have also been deposited in the UK NERC National Geoscience Data Centre (https://doi.org/10.5285/d7094583-4564-4651-85ea-d19e6261a31e).

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

## Acknowledgements

This research has been funded by the Strategic Priority Research Program of the Chinese Academy of Sciences (XDA14010103 to Y.W.), the Natural Environment Research Council of UK (Grant Ref: NE/T004452/1 to Z.Z.), the National Natural Science Foundation of China (Grant No. 41872162, 42141022 to S.Q., and 42203054 to C.C.), and the Guangdong Basic and Applied Basic Research Foundation (Grant No. 2022A1515011823 to C.C.). C.C. acknowledges China Scholarship Council (CSC) for financial support (File No. 201904910306). We thank D. Nelson, S. Davis, S. Huang, and Q. Wang for their technical support in laboratory analysis. This is contribution No.IS-3269 from GIGCAS.

## Author contributions

Z.Z., Y.W., S.Q., and C.C. designed the study and collected the samples. C.C. W.Z. Y.W., and Z.Z. analyzed the samples. The manuscript and figures were drafted by C.C., Z.Z., and G.H. with contributions from all authors, specifically including advice on petroleum geology from S.Q. and methane geochemistry from P.W.

## Competing interests

The authors declare no competing interests.
