## [Peer Review File · Nature Communications]

High temperature methane emissions from Large Igneous Provinces as contributors to late Permian mass extinctionsReviewer #1 (Remarks to the Author):

Review of Chen et al. - High temperature methane emissions from the large igneous provinces as contributors to late Permian mass extinctions (NCOMMS-22-17325-T)

Major comments:

Chen et al. report clumped methane measurements from 9 CH₄ samples and noble gas data from ~20 samples. They compute clumped methane temps of 249-17/+19–256-20/+22°C, which is on the high end of what might be expected for a dry methane gas system. They state the reservoir estimates are up to 220C, so while these values are high, they are not so high that they necessitate a plume or LIP contribution. The authors go on to claim a “strong mantle influence” on gases however it is only apparent in a subset of Z2dn samples have elevated He isotopes, as far as I can tell, the rest of the samples are strongly radiogenic. They later explain that crustal noble gas signatures could be explained by underlying thick carbonate or shale strata, and that only the samples with basement underneath show any evidence for mantle influence. This is hard to reconcile with their larger model of massive CH₄ release. Those strata retained their radiogenic parents (U, Th, K) despite the LIP event releasing huge amounts of CH₄? I think this could be explained in more detail, maybe I missed it.

The authors claim these geochemical data “verify that oil-cracked CH₄ and pyrobitumen are by-products within the reservoirs”, associated with hydrothermal activity and enhanced heating of the ELIP, which as far as I can tell might be true (admittedly I’m a geochemist, so it’s admittedly hard for me to evaluate the significance of the honeycomb structures within pyrobitumen). The authors then go on to estimate the volume of oil-cracked CH₄ induced by the ELIP and argue that CH₄ emissions would have been sufficient to initiate global warming prior to the end of the Permian era. If their calculations are accurate it would indeed be a huge amount of methane release, but I found it impossible to follow what they calculated as the equations were either missing from the SOM table or I missed them (again this could be my fault). The authors also suggest that similar emissions from oil-cracked CH₄ associated with the Siberian Traps Large Igneous Province (STLIP) may also have contributed to the end-Permian mass extinction (EPME).

The authors are highly respected in their field and known to produce excellent data and robust interpretations. I found the manuscript to be interesting and well written, but am not convinced that the conclusions are fully supported by the data. I don’t see any “smoking guns” in the geochemical data to support their theory but I cannot definitively say they are wrong either. I couldn’t reproduce the calculations and I think the authors could be more transparent about how they did that (walk the reader through this critical calc). I think the paper should be published as the data are highly interesting but I’d like to see the story more strongly supported by the data and a better presentation of how the calculations were done. I’ll leave the decision to the Nat Comm editors as to if this is the right venue for the paper.

Line by line comments:

Because the authors did not provide line numbers or page numbers it is very hard to provide detailed line by line comments, although I made several notes.

In the intro it would be good to explicitly state that ch₄ is a much stronger greenhouse gas: something like Methane is a powerful greenhouses gas with a 100-year global warming potential 28-34 times that of CO₂. It seems later when they do their calculation they assume 28x stronger greenhouse gas vs co₂, but again it would be good to state the assumptions so the reader can also calculate.

At the end of the intro it seems like the results could be stated more explicitly, rather than this general statement “we quantified methane generation”. In such a short form

paper it is necessary to get to the point quicker, in my opinion.

Fig 1 caption, write out nine rather than starting a sentence with 9

Top of page 4 – say hydrothermal activity, not activities

On page 5 under the constraining methane formation temp section: I'd argue 250 clumped isotope temp is slightly above the 220 res estimate not significantly above. Nothing about this temp screams plume or LIP influence to me, but it doesn't refute this theory either, which really seems based on the honeycomb micropores and carbon foams in the pyrobitumen, but again this isn't my realm of expertise.

Fig 2b (And throughout) do these reservoirs have other names? Seems odd to call them z2dn and e1l, but up to the authors I guess. Are there two subgroups for z2dn, the table seems to indicate yes.

Page 7 – when you discuss He and Ar isotopes, its interesting that the high $^3\text{He}/^4\text{He}$ samples have the most air like Ar isotopes. This seems like it should be discussed... are all other samples just being overprinted by radiogenic He and Ar from the crust? What differentiates the z2dn subunit with high $^3\text{He}/^4\text{He}$? Compartmentalization? Migration distance? Underlying lithology?

Many groups, including the senior author (Zhou) have modelled migration distances using air derived noble gases (^{36}Ar , ^{20}Ne). Can some insight be gained about the history of the gases using such approaches?? Do the gases with the largest mantle anomalies show evidence for long migration distances? I'd guess yes based on table s3 that shows the samples with high $^3\text{He}/^4\text{He}$ also have much higher ^{20}Ne and ^{36}Ar , meaning that gas was in contact with a lot of water potentially (during migration). This is touched on at the bottom of page 7 where the authors say that the radiogenic signals could have to do with thick carbonate systems underneath the reservoirs, and the lack of such depositional systems underneath the z2dn res. How for instance is this reconciled with the much higher ^{20}Ne and ^{36}Ar ?

Fig.3a – if it is a plume induced plume then why mix with a SCLM endmember? I don't know of any studies of He isotopes in the Emeishan that show evidence for plume, but in Siberia Basu et al., 1995 showed evidence for plume.

3b – I don't understand how this mixing line is drawn... why doesn't it go to the crustal production value? How would migration or dissolution affect these isotope ratios? Why do the samples with high $^3\text{He}/^4\text{He}$ have 40/4 higher than mantle production 40/4 ratios? This seems important but is not explained. Again, does this have to do with gas movement in the crust and/or nearby lithology?

Bottom of pg 9 you discuss total methane generation but I don't follow exactly what you did. The table nicely lays out the parameters used in the equations but the equations 1 and 2 that you refer to seem to not exist or be referring to how clumped isotopes are defined. This is a key point of the paper, and I don't doubt your result, just think you should explain what you did in more detail (ideally in the main text).

Reviewer #3 (Remarks to the Author):

In this manuscript the Authors investigate the CH_4 emissions from the Emeishan LIP as likely responsible of the end-Guadalupian mass extinction. Similar emissions via cracking of pre-existing oil reservoirs are suggested also for the Siberian Traps LIP, indicating a potential contribution of this process to the end-Permian mass extinction too. Natural gas samples from the Sinian-Cambrian reservoirs in the Anyue gas field (central Sichuan basin) were analysed for the clumped isotopes of CH_4 in order to obtain its formation temperature, and for the isotopic signature of noble gases in order

to track the mantle contribution during the activity of the Emeishan LIP. Furthermore, the Authors carried out a numerical modelling to reconstruct the basin evolution and the hydrocarbon generation, estimating the amount of CH₄ produced via oil pyrolysis within the reservoirs and emitted in the palaeo-atmosphere during the activity of this LIP.

Since the manuscript has been submitted without line numbers, all my comments are reported on the .pdf file of the submitted manuscript. However, I specify that I am not expert in isotope geochemistry and I have no experience with noble gases. Thus, I would recommend another Reviewer to check the technical details of these analytical procedures.

The Authors used a very interesting approach to constrain part of the CH₄ emissions induced by the emplacement of the Emeishan LIP, reaching interesting conclusions that may apply for many other LIPs. The mechanism of oil cracking to produce CH₄ (and pyrobitumen as by-product) highlight the importance of both hydrothermal activity and thermal anomaly induced by LIPs to further contribute to their degassing activity, besides volcanic emissions, thermal metamorphism and coal combustion.

I would require some minor changes, as reported in my comments in the .pdf file. I would appreciate also that all the data would be formatted according to the scientific notation in the main text, and that the final calculations about equivalent CO₂ amounts of CH₄ would be explained more properly, as reported in my specific comments. This manuscript presents new interesting data and enriches the understanding of the degassing activity of LIPs in general, thus I recommend the publication of this manuscript after minor revision.

In their review of the first version of this manuscript, reviewer #3 added some comments to the manuscript file. These comments were forwarded to the authors, who replied as included in this Peer Review File.

Response to Reviewers' Comments

We have made significant revision to the manuscript initially submitted. Please see below point-by-point response to the reviewers' comments and concerns. Line numbers in our response to reviews' comments indicate where changes are made in the revised version of the manuscript.

Comments from Reviewer #1

Major comments:

Comment 1: Chen et al. report clumped methane measurements from 9 CH₄ samples and noble gas data from ~20 samples. They compute clumped methane temps of 249-17/+19–256-20/+22°C, which is on the high end of what might be expected for a dry methane gas system. They state the reservoir estimates are up to 220°C, so while these values are high, they are not so high that they necessitate a plume or LIP contribution. The authors go on to claim a “strong mantle influence” on gases however it is only apparent in a subset of Z₂dn samples have elevated He isotopes, as far as I can tell, the rest of the samples are strongly radiogenic. They later explain that crustal noble gas signatures could be explained by underlying thick carbonate or shale strata, and that only the samples with basement underneath show any evidence for mantle influence. This is hard to reconcile with their larger model of massive CH₄ release. Those strata retained their radiogenic parents (U, Th, K) despite the LIP event releasing huge amounts of CH₄? I think this could be explained in more detail, maybe I missed it.

Response 1: There are two concerns here in this comment. The first is whether the calculated clumped methane temperatures are high enough to necessitate a plume or LIP contribution. The second is why elevated ³He/⁴He ratios were only found in the Z₂dn samples and whether large methane releases from Z₂dn and C₁l and subsequent ingrowth of radiogenic noble gases would affect their ³He/⁴He ratios.

Regarding the first concern, our calculated clumped methane formation temperatures are significantly higher the geological temperature at the maximum burial depth during Late Cretaceous which is less than 220°C. This indicates that CH₄ in the Anyue gas field is unlikely generated by the burial process. In addition, the dryness coefficients of gas samples ($C_1/\Sigma C_{2-5} = 583\sim 3019$) require a gas formation temperature beyond 250°C. Also, petrological evidence from pyrobitumen suggests that the reservoirs had undergone a coking process by hydrothermal fluid invasion associated with the ELIP. We have added a discussion on this comment in the second and third paragraphs in the subsection “Constraining methane formation temperatures in the Anyue field using $\Delta^{13}\text{CH}_3\text{D}$ values” in “Results and Discussion”. All evidence suggests that the Anyue gas field has experienced rapid oil cracking, gas expansion, and escape processes induced by the ELIP. Gas generation by burial played only a minor role in the Anyue region.

For the second concern, we explained in the revised manuscript that elevated $^3\text{He}/^4\text{He}$ ratios were only found in the Z_2dn samples due to sequence stratigraphical characteristics of the Anyue gas field. The Z_2dn reservoirs were located at the bottom of the strata close to the basement where the mantle influence occurred, and there were thick shale layers separating the Z_2dn reservoirs from the C_{1l} reservoirs above implying isolation between them. Degassing 250Ma ago followed by later generation of ^4He would give lower $^3\text{He}/^4\text{He}$ today. Samples with the highest $^3\text{He}/^4\text{He}$ ratios do not have the highest ^3He concentrations, which would imply variable amounts of radiogenic ^4He generated subsequently, variable amounts of degassing early and/or variable amounts of ^3He from mantle in the first place. However, radiogenic ^4He does not cover up the mantle impact in gas generation in the Anyue gas field.

Comment 2: The authors claim these geochemical data “verify that oil-cracked CH_4 and pyrobitumen are by-products within the reservoirs”, associated with hydrothermal activity and enhanced heating of the ELIP, which as far as I can tell might be true (admittedly I’m a geochemist, so it’s admittedly hard for me to evaluate the significance of the honeycomb structures within pyrobitumen). The authors then go on to estimate the volume of oil-cracked CH_4 induced by the ELIP and argue that CH_4 emissions would have been sufficient to initiate global warming prior to the end of the Permian era. If their calculations are accurate it would indeed be a huge amount of methane release, but I found it impossible to follow what they calculated as the equations were either missing from the SOM table or I missed them (again this could be my fault). The authors also suggest that similar emissions from oil-cracked CH_4 associated with the Siberian Traps Large Igneous Province (STLIP) may also have contributed to the end-Permian mass extinction (EPME).

Response 2: Thank you for pointing this out. We have included in the manuscript detailed explanation on how the volume of oil-cracked CH_4 induced by the ELIP is calculated, including equations and parameters used. Please see subsection “Methane emission model” in “Methods”.

Comment 3: The authors are highly respected in their field and known to produce excellent data and robust interpretations. I found the manuscript to be interesting and well written, but am not convinced that the conclusions are fully supported by the data. I don’t see any “smoking guns” in the geochemical data to support their theory but I cannot definitively say they are wrong either. I couldn’t reproduce the calculations and I think the authors could be more transparent about how they did that (walk the reader through this critical calc). I think the paper should be published as the data are highly interesting but I’d like to see the story more strongly supported by the data and a better presentation of how the calculations were done. I’ll leave the decision to the Nat Comm editors as to if this is the right venue for the paper.

Response 3: Thank you for this comment. We have revised the manuscript to include details on all our calculations and provided further discussion according to your and other reviewer’s comments. We believe our arguments in the revised manuscript are now strongly supported by the data, calculation, modelling, and discussion.

Line by line comments:

Comment 4: In the intro it would be good to explicitly state that CH₄ is a much stronger greenhouse gas: something like Methane is a powerful greenhouse gas with a 100-year global warming potential 28-34 times that of CO₂. It seems later when they do their calculation they assume 28x stronger greenhouse gas vs CO₂, but again it would be good to state the assumptions so the reader can also calculate.

Response 4: We have added a sentence in the second paragraph of “Introduction” to make this statement. Also, it has been emphasized when discussing methane emission in the subsection “The extent of methane production and emission induced by ELIP” in “Results and Discussion”.

Comment 5: At the end of the intro it seems like the results could be stated more explicitly, rather than this general statement “we quantified methane generation”. In such a short form paper it is necessary to get to the point quicker, in my opinion.

Response 5: Agree. We have included result for methane emission from the Anyue gas field more explicitly at the end of the Introduction.

Comment 6: Fig 1 caption, write out nine rather than starting a sentence with 9

Response 6: We have made this change.

Comment 7: Top of page 4 – say hydrothermal activity, not activities

Response 7: We have made this correction.

Comment 8: On page 5 under the constraining methane formation temp section: I’d argue 250 clumped isotope temp is slightly above the 220 res estimate not significantly above. Nothing about this temp screams plume or LIP influence to me, but it doesn’t refute this theory either, which really seems based on the honeycomb micropores and carbon foams in the pyrobitumen, but again this isn’t my realm of expertise.

Response 8: You have raised an important point here. However, we considered our clumped methane temperature of 250°C was significantly higher than reservoir temperature of up to 220°C estimated by maximum burial depth, because small temperature difference would generate completely different end products in organic reactions after considering the combined effect of temperature and time. This has been demonstrated in gas dryness coefficients and petrological characteristics in the pyrobitumen. We have made clarification and discussion on this point in the revised manuscript. Please see the second and third paragraphs of “Constraining methane formation temperatures in the Anyue field using $\Delta^{13}\text{CH}_3\text{D}$ values” in “Results and Discussion”.

Comment 9: Fig 2b (And throughout) do these reservoirs have other names? Seems odd to call them z2dn and e11, but up to the authors I guess. Are there two subgroups

for z2dn, the table seems to indicate yes.

Response 9: Thank you for your suggestion. We would like to keep using C_{1l} and Z_2dn as reservoir names, because they have been used in many literatures. The numbers in these names (i.e., 1, 2, and 3) represent the early, middle, and late period in which strata are deposited. The numbers '1' and '2' in this manuscript indicate early Cambrian and middle Sinian sedimentary strata while the symbols are conventional usage in many literatures (e.g., Liu et al., 2017). As for subgroups of Z_2dn -G1 and G2, they do not correspond to different reservoirs but are classification groups according to the values of $^3\text{He}/^4\text{He}$ ratios, taking 1Ra as the separation divider. They facilitate our description and discussion. We have made this clear in the revised manuscript.

Comment 10: Page 7 – when you discuss He and Ar isotopes, its interesting that the high $^3\text{He}/^4\text{He}$ samples have the most air like Ar isotopes. This seems like it should be discussed... are all other samples just being overprinted by radiogenic He and Ar from the crust? What differentiates the z2dn subunit with high $^3\text{He}/^4\text{He}$? Compartmentalization? Migration distance? Underlying lithology?

Response 10: We have made a discussion in the manuscript about the high $^3\text{He}/^4\text{He}$ ratios and their link to air like $^{40}\text{Ar}/^{36}\text{Ar}$ ratios in the Sinian Z_2dn -G1 samples. High $^3\text{He}/^4\text{He}$ ratios are due to the deposition environment of Z_2dn . It is located close to the basement of the basin and directly influenced by mantle activity which provides the source of high $^3\text{He}/^4\text{He}$ ratios. Other samples with low $^3\text{He}/^4\text{He}$ ratios are from the Cambrian reservoirs (C_{1l}) above the Sinian Z_2dn reservoirs. They are enclosed in thick shale layers (Fig.1c), which prevents the contact with the hydrothermal fluids with high $^3\text{He}/^4\text{He}$ ratios. Air like $^{40}\text{Ar}/^{36}\text{Ar}$ ratios in the Z_2dn reservoirs indicated that they are less influenced by radiogenic Ar than other reservoirs, possibly by underlying lithology.

Comment 11: Many groups, including the senior author (Zhou) have modelled migration distances using air derived noble gases (^{36}Ar , ^{20}Ne). Can some insight be gained about the history of the gases using such approaches?? Do the gases with the largest mantle anomalies show evidence for long migration distances? I'd guess yes based on table s3 that shows the samples with high $^3\text{He}/^4\text{He}$ also have much higher ^{20}Ne and ^{36}Ar , meaning that gas was in contact with a lot of water potentially (during migration). This is touched on at the bottom of page 7 where the authors say that the radiogenic signals could have to do with thick carbonate systems underneath the reservoirs, and the lack of such depositional systems underneath the z2dn res. How for instance is this reconciled with the much higher ^{20}Ne and ^{36}Ar ?

Response 11: We have investigated further into using noble gas isotopes for tracing the gas migration history in the Anyue gas field. $^{20}\text{Ne}/^{36}\text{Ar}$ ratios in the Cambrian samples (C_{1l}) range between 0.14 and 0.25. They are close to the $^{20}\text{Ne}/^{36}\text{Ar}$ value of ~ 0.16 in air saturated water (ASW), demonstrating a near total degassing of water derived noble gas isotopes in the Cambrian reservoirs. However, all $^{20}\text{Ne}/^{36}\text{Ar}$ ratios in the Sinian samples (Z_2dn) range between 0.33 and 0.91 except an extreme value of 1.86. They are deviated from the ASW value, which can potentially be used for quantifying gas migration history. In a plot of $^{20}\text{Ne}/^{36}\text{Ar}$ vs. $^{40}\text{Ar}^*/^4\text{He}$ (please see below), water-derived

^{20}Ne and ^{36}Ar isotopes and radiogenic ^{40}Ar and ^4He isotopes in the Z_2dn -G1 and Z_2dn -G2 reservoirs demonstrate different fractionation patterns. This suggests that gases in the Z_2dn -G1 and Z_2dn -G2 reservoirs might have experienced different migration history. In addition, high ^{20}Ne and ^{36}Ar concentrations in the Z_2dn samples might suggest a possible meteoric water recharge associated with the Z_2dn reservoir. Therefore, while ^{20}Ne and ^{36}Ar isotopes can be used for tracing gas migration history, gas-groundwater interaction, and quantifying gas water volume ratios, etc., they have preserved information from events other than the ELIP. On the contrary, $^3\text{He}/^4\text{He}$ signatures from the ELIP can be well preserved in the gas samples and not affected significantly by other geological processes. As our focus of this paper is on tracing the impact of the ELIP on methane generation and emission in the Anyue field, we wish to use elevated $^3\text{He}/^4\text{He}$ to indicate the presence of a mantle plume and associated heat and show that variable noble gas concentrations indicate variable but extensive degassing, to support the independent evidence for methane degassing which is presented elsewhere in the manuscript. We will return to the noble gas systematics in a subsequent paper that focuses on the relationships between noble gas compositions in these strata, because there is a complex interplay between migration, water interaction, mantle fluxes and variable degassing beyond the scope of this paper. Therefore, we have decided not to include a full discussion on noble gas systematics in this manuscript.

Comment 12: Fig.3a – if it is a plume induced plume then why mix with a SCLM endmember? I don't know of any studies of He isotopes in the Emeishan that show

evidence for plume, but in Siberia Basu et al., 1995 showed evidence for plume.

Response 12: There are no previous studies reporting such high $^3\text{He}/^4\text{He}$ ratios in the Sichuan Basin as a stable cratonic basin. We suggest that they are related to the ELIP derived from the Emeishan mantle plume (EMP) with this new evidence. Previous research proposed that the ELIP was a mafic continental large igneous province developed in a typical non-rifted continental margin. It then formed volcanic successions, such as large amounts of continental flood basalts (Xu et al., 2004). The presence of picritic rocks and thick piles of flood basalts could be linked to high temperature thermal regime, but there was uncertainty as to whether these magmas were derived from the subcontinental lithospheric mantle (SCLM) or sub-lithospheric mantle (i.e., asthenosphere or mantle plume) source or both. It was showed that some rocks were contaminated by crustal contribution and there was a SCLM component in the system (Shellnutt, 2014). Furthermore, helium isotope systematics have given good constraints on mid-ocean ridge basalts (MORB) (i.e., depleted upper mantle; $R/Ra=8\pm 1$) and SCLM ($R/Ra=6.1\pm 0.9$) endmembers, but values of helium ratios in ocean island basalts (OIB) vary in a large range ($R/Ra=4.00\sim 42.9$). Among all known helium endmembers, the SCLM endmember was chosen for this study because the ELIP was continental, which should show a relatively homogeneous and more radiogenic characteristic than the endmember from the MORB source (Gautheron and Moreira, 2002). We have added the explanation above in the revised manuscript. Choosing a different endmember value does not affect the conclusions of this manuscript.

As for the paper Basu et al., 1995, authors showed the origin of plume with high- ^3He ($R/Ra=12.7Ra$) tested in olivine phenocrysts in the Siberian Trap Large Igneous Province (STLIP) and claimed an interaction of the high- ^3He plume with a suboceanic-type upper mantle beneath the Siberia, which was different from the ELIP with a continental origin.

Comment 13: 3b – I don't understand how this mixing line is drawn... why doesn't it go to the crustal production value? How would migration or dissolution affect these isotope ratios? Why do the samples with high $^3\text{He}/^4\text{He}$ have $^{40}\text{Ar}/^4\text{He}$ higher than mantle production $^{40}\text{Ar}/^4\text{He}$ ratios? This seems important but is not explained. Again, does this have to do with gas movement in the crust and/or nearby lithology?

Response 13: Thank you for this good point. The mixing line in Fig. 3b is an extrapolated mixing line between the crust and mantle, which has been defined by Stuart et al. (1995) in unfractionated cases that measured in Dae Hwa (South Korea) W-Mo deposit fluid inclusions (Stuart et al., 1995; Ballentine et al., 2002). This clarification has been added in the caption of Fig. 3 and main text.

We used, in the mixing line, the mantle endmember value of $^{40}\text{Ar}^*/^4\text{He}=0.69\pm 0.06$, which is typical unfractionated samples from the mantle (Graham, 2002), and the crustal endmember value of $^{40}\text{Ar}^*/^4\text{He}=0.007$ (Ballentine et al., 2002). The crustal production ratio of $^{40}\text{Ar}^*/^4\text{He}\sim 0.2$ in the figure is a production weighted average calculated by using the lower, middle, and upper crustal element compositions (Ballentine and Burnard, 2002).

Samples with high $^3\text{He}/^4\text{He}$ ratios also have $^{40}\text{Ar}^*/^4\text{He}$ ratios higher than mantle production $^{40}\text{Ar}/^4\text{He}$ ratios. $^{40}\text{Ar}^*/^4\text{He}$ fractionation could be explained by several mechanisms. The first is the localized ^{40}Ar and ^4He production, which could produce either high or low $^{40}\text{Ar}/^4\text{He}$ ratios. The second is gas migration, diffusion and adsorption processes which could cause $^{40}\text{Ar}/^4\text{He}$ ratios lower than the production ratio. The third one is related to gas and water interaction, which could make $^{40}\text{Ar}/^4\text{He}$ ratios in the gas phase lower than the production ratio due to relatively higher ^{40}Ar solubility than ^4He solubility. We suggested in the revised manuscript that, due to heat from ELIP, samples with high $^3\text{He}/^4\text{He}$ ratios also have high $^{40}\text{Ar}^*/^4\text{He}$ ratios, because high temperature would release $^{40}\text{Ar}^*$ trapped in minerals. Therefore, migration and dissolution would affect $^{40}\text{Ar}^*/^4\text{He}$ ratios, and cause the ratios in the gas phase smaller than initial values. These processes can be discounted. However, these processes do not affect $^3\text{He}/^4\text{He}$ ratios. They can be used as indicator for mantle involvement reliably.

Z₂dn-G1 samples which have high $^3\text{He}/^4\text{He}$ ratios also have high $^{40}\text{Ar}^*/^4\text{He}$ ratios. We suggest that this is also due to the influence from the hydrothermal fluid. High temperature would release high amount of $^{40}\text{Ar}^*$ trapped in minerals. However, *Z₂dn-G2* samples are less affected by the hydrothermal fluid with low $^{40}\text{Ar}^*/^4\text{He}$ ratios and scattered $^{20}\text{Ne}/^{36}\text{Ar}$ ratios, as discussed in our response to comment 11. We have included this discussion in our revised manuscript.

Comment 14: Bottom of pg 9 you discuss total methane generation but I don't follow exactly what you did. The table nicely lays out the parameters used in the equations but the equations 1 and 2 that you refer to seem to not exist or be referring to how clumped isotopes are defined. This is a key point of the paper, and I don't doubt your result, just think you should explain what you did in more detail (ideally in the main text).

Response 14: Thank you for pointing this out. We have included in the revised manuscript detailed explanation on how the volume of oil-cracked CH_4 induced by the ELIP is calculated, including equations and parameters used. Please see subsection "methane emission model" in "methods".

Comments from Reviewer #2

Comment 1: The literature review is not completed, the authors missed the major studies on different causes of PT boundary mass extinction especially recent publication on organic matter combustion and also a latest paper on the same issue of the paleo-oil reservoir pyrolysis in the study area.

[1] Grasby, S.E., Sanei, H., Beauchamp, B., 2011. Catastrophic dispersion of coal fly ash into oceans during the latest Permian extinction. *Nat. Geosci.* 4, 104-107.

<https://doi.org/10.1038/ngeo1069>

[2] Chengyu Yang, Meijun Li, Zhiyong Ni, Tieguan Wang, Nansheng Qiu, Ronghui Fang, Long Wen, Paleo-oil reservoir pyrolysis and gas release in the Yangtze Block

imply an alternative mechanism for the Late Permian Crisis, *Geoscience Frontiers*, Volume 13, Issue 2, 2022, 101324, <https://doi.org/10.1016/j.gsf.2021.101324>.

Response 1: Thank you for pointing out the deficiency in our literature review. Both papers are important for our manuscript and worth citing, although we were not aware of Yang et al. (2022) at the time when we completed our manuscript. We have included them in the Introduction of our revised manuscript.

Comment 2: The introduction section: “there is no evidence to directly link the massive generation of oil-cracked CH₄ within the ELIP, to the Emeishan mantle plume (EMP), Therefore, it remains unclear whether the ELIP could have acted as a “coking furnace” for promoting the massive generation of oil-cracked CH₄ and pyrobitumen underground” If the authors refer to the paper by Yang et al., 2022, this paragraph should be rewritten. Yang et al., have proposed the possible source of greenhouse gas and even hydrogen sulfide from pyrolysis of petroleum in pre-Permian reservoirs.

Response 2: Yang et al. (2022) suggested that venting of carbon gases (CH₄ and CO₂) and even halocarbons might have been released into the atmosphere through pipes in the Sichuan Basin, China during the Late Permian. However, this work was based on indirect evidence in the petrological recordings to infer the possibility of gas emissions. They did not quantify the extensiveness of the pyrolysis process and had insufficient evidence of its strong correlation with the LIPs. In addition, they underestimated contribution of global warming from CH₄ in the absence of well-established CH₄ production-emission model. Therefore, we consider our statement is still valid. Nevertheless, we have made an introduction to Yang et al. (2022), and revised our manuscript.

Comment 3: This study provided more evidence using some up-to-date technology, e.g., clumped methane isotope measurements to further confirm the high methane formation temperature, as well as the mantle-derived signatures of noble gases.

Response 3: Yes, we consider information derived from these geochemical techniques direct evidence for high temperature methane emissions, which is different from previous studies.

Comment 4: I understand that linking with the largest mass extinction might sound exciting, but please give a consideration of the G-L mass extinction event. The current data cannot support a clear link of the Emeishan LIP with the P-T extinction. If the authors insist on linking with the P-T, please at least mention the possibility of the G-L event.

Response 4: Thank you for this great point. We have mentioned the G-L event and linked it with the ELIP throughout the manuscript. We proposed that the same mechanism for high temperature methane emissions might work in STLIP during the P-T extinction in our discussion.

Comment 5: Another main point is the timing. The author should clarify the timing of the Eishen LIP (252 Ma or 260 Ma), the timing of paleo-oil reservoir forming and the timing of oil cracking.

Response 5: This is again another great point. We have clarified the timing of the ELIP, paleo-oil reservoir forming, and oil cracking, in Lines 33-37, 81-83 and 274-276, respectively.

Comment 6: Type-I kerogen was used in the basin modelling, do you have more data to backup your statement?

Response 6: Yes, we have data to support our statement. For example, the Qiongzhusi Formation (C_{1q}) in the Sichuan Basin has been well studied, which is regarded as one of the best source rocks in the basin and even in South China. It has a very high abundance of organic matter with TOC values even larger than 6% (averaged at 2.84%). Organic matters in the Qiongzhusi Formation were mainly bacteria and algae, as its depositional setting was a reducing environment with moderate salinity. In such a depositional environment, benthos such as cnidarians, arthropods, mollusks, and macroalgae were common, especially poriferans. Therefore, the organic matter in the source rocks is mainly of the sapropel type, which indicates Type-I kerogen (Zou et al., 2014; Zhu et al., 2015; Liu et al., 2017). We have made clarification on this point in the Methods and supplementary Fig. S3.

Comments from Reviewer #3

Major comments

I would require some minor changes, as reported in my comments in the .pdf file. I would appreciate also that all the data would be formatted according to the scientific notation in the main text, and that the final calculations about equivalent CO₂ amounts of CH₄ would be explained more properly, as reported in my specific comments.

Response:

We have made revision following all suggestions in the .pdf file, please see below our point-by-point response. All the data have been formatted to the scientific notation in the main text. The final calculations about equivalent CO₂ amounts of CH₄ have been explained in more detail with parameters and equations provided.

Comments 1-42 from reviewer #3 in the annotated .pdf file

Comment 1: Here, I suggest to use the capital letter for "Large Igneous Provinces".

Response 1: We have used the capital letter for "Large Igneous Provinces".

Comment 2: Do you mean "Ma"?

Response 2: Yes. Myrs has been changed into Ma throughout the manuscript.

Comment 3: Here, I suggest again to use the capital letter for "Large Igneous Provinces".

Response 3: This change has been made.

Comment 4: Is this referred to each event or to the two overall? Clarify, please.

Response 4: We have made it clear in the manuscript it is referred to the two events overall.

Comment 5: I suggest to rephrase as "clumped isotopes for methane and isotope tracers for noble gas".

Response 5: We have taken this suggestion and revised the manuscript.

Comment 6: Check the spelling of "Micang Mountain" here and "Continent/Continental" in the stratigraphic column.

Response 6: We have made the corrections. Thanks.

Comment 7: I think that the geographic distribution of the ELIP is shown in the small inset of panel (a), but it is unclear. And what does "guide line" indicate? Clarify, please.

Response 7: We have made it much clearer in Fig. 1a to show the geographic distribution of the ELIP. We have clarified that the "guide line" can help locate the study area in this and the following figures (Fig. 1a, Fig. 1b, and Fig. 4b). We have also made it clear what the "guide line" means in Fig. 4b.

Comment 8: Unclear. Rephrase, please.

Response 8: We have rephrased the Fig. 1b and made it much clearer.

Comment 9: Do the red dots indicate samples from 9 different gas reservoirs? Clarify, please.

Response 9: We have made it clear that nine red dots are samples collected for clumped methane and noble gas analysis. They are from different wellheads within the gas field producing gases from both Cambrian and Sinian reservoirs.

Comment 10: "evolution" is probably better here. Alternatively, provide more details, please.

Response 10: We have used the word "evolution" and provided more details about the it in the caption for Figure 1c.

Comment 11: I suggest to rephrase as "Magmas from the EMP".

Response 11: We have made revision following this suggestion. Due to formatting requirements of Nature Communications, we have moved the geological setting into the Introduction. Therefore, we have made some minor changes of texts. Please see the revised Introduction.

Comment 12: Are you referring to the small inset of panel (a)?

Response 12: Yes. Please see the revised Introduction.

Comment 13: Be consistent in using "myrs" or "Ma".

Response 13: We have used “Ma” throughout the revised manuscript.

Comment 14: This is already part of the "Results" section.

Response 14: According to your suggestion and the formatting requirements of Nature Communications, we have moved this part into subsection Gas composition and geochemical characteristics in Results and Discussion.

Comment 15-17: Do you mean "from 583 to 3019"? /Specify if this is a range. /Specify again if this is a range.

Comments 15-17: We have rewritten them as “with a range from...to...”. Thanks for your suggestion.

Comment 18: Be consistent in putting the minimum or maximum before.

Response 18: We have made it consistent to put all the maximum values before minimum values.

Comment 19: I would add "likely" here, since just an interpretation. If not excluded by isotope data on H₂S and CO₂, these gases may also have a magmatic origin.

Response 19: We have added “likely” in the revised manuscript.

Comment 20: All gases from these two reservoirs or from the entire basin? Clarify, please.

Response 20: We have clarified that gases are from the two reservoirs.

Comment 21: Specify which equilibrium conditions you mean here.

Response 21: We have emphasized in the manuscript that gases have reached equilibrium in terms of clumped methane isotopes.

Comment 22: How could you exclude oil-cracking at the maximum reservoir temperature according to your model (i.e., up to 220 °C)?

Response 22: We excluded oil-cracking at the maximum reservoir temperature (up to 220°C) by gas composition in samples, petrological evidence, and our calculated clumped methane formation temperatures. The dryness coefficients of gas samples ($C_1/\Sigma C_{2-5} = 583\sim 3019$) require a gas formation temperature beyond 250°C. Petrological evidence from pyrobitumen also suggests that the reservoirs had undergone a coking process by hydrothermal fluid invasion associated with the ELIP. More importantly, the calculated clumped methane formation temperatures are significantly higher than the geological temperature at the maximum burial depth during Late Cretaceous which is less than 220°C. This indicates that CH₄ in the Anyue gas field is unlikely generated by the burial process. We have made a discussion on this comment in the subsection “Constraining methane formation temperatures in the Anyue field using $\Delta^{13}CH_3D$ values” in “Results and Discussion”. All evidence suggests that the Anyue gas field has experienced rapid oil cracking, gas expansion, and escape processes induced by the

ELIP. Gas generation by burial only played a minor role in the Anyue region.

Comment 23: Do you mean "fluid inclusions within quartz"? And are they in the same gas reservoirs? Clarify, please.

Response 23: Yes, the "fluid inclusions within quartz" we discussed in the manuscript are in the same gas reservoirs in the Central Sichuan Basin. This has been clarified in the text.

Comment 24: Convincing and very interesting!

Response 24: Thanks.

Comment 25: What do green and red colours indicate? Specify here or in the caption, please.

Response 25: We have specified the meaning of red and green patches in Fig. 2a in its caption.

Comment 26: Shouldn't it be gray?

Response 26: Yes, we have made this correction. Thanks.

Comment 27: Keep the same number of significant figures for the same type of data.

Response 27: We have followed this suggestion throughout the revised manuscript.

Comment 28: I would add "likely" here, since just an interpretation.

Response 28: We have included the word "likely"

Comment 29: $9.96 \times 10^{(6)}$

Response 29: We have followed this suggestion.

Comment 30: How is this mixing line defined?

Response 30: The mixing line between the crust and mantle endmembers was defined by Stuart et al. (1995) using unfractionated cases that measured in Dae Hwa (South Korea) W-Mo deposit fluid inclusions. The mantle $^{40}\text{Ar}^*/^4\text{He}=0.69\pm 0.06$ is typical of unfractionated samples from the mantle. In contrast, the crustal $^{40}\text{Ar}^*/^4\text{He}=0.007$ is far lower than the crustal production ratio of approximately 0.2, typically representing a fluid derived from shallow cool regions of the crust, in which crustal fluids often mix with a He rich endmember due to preferential addition of ^4He to the gas phase. We have added this explanation in the caption for Fig.3.

Comment 31: I would use the verb "suggests" more than "demonstrates" here.

Response 31: We have replaced "demonstrates" with "suggests".

Comment 32: What do you mean with "other solid minerals"? Minerals are solid by

definition.

Response 32: We meant “other hydrothermal-related minerals”. We have made this correction.

Comment 33: Is there any evidence of hydrothermal pipes or similar structures?

Response 33: Yes, we have added a discussion on this point.

Comment 34: Are you referring to the previously cited fluid inclusions within quartz? Clarify, please.

Response 34: In addition to the fluid inclusions within quartz, we cited other literature to support this argument. We have clarified this in the manuscript.

Comment 35: This equivalence is wrong, but I read from the Supplementary Information that you mean "equivalent CO₂ emission" as defined by the IPCC report (2013): "the global warming potential of CH₄ is 28 times of CO₂ in a 100-year period". This has to be specified also in the main text. Clarify, please.

Response 35: We have made clarification in the manuscript.

Comment 36: This number is from the previous equivalence. However, this should be better explained since you consider 28 times the original amount of CH₄!!!

Response 36: We have explained this point in more detail in the manuscript following your suggestion. Thanks.

Comment 37: Great figure!

Response 37: Thanks.

Comment 38: I don't get the sequentiality in this statement. I think that the Sinian reservoirs maintained the information of mantle-derived gases better than the Cambrian reservoirs just because of their lower stratigraphic position, independently from the following (from late Cretaceous) tectonic evolution. Clarify, please.

Response 38: This is a great point, and we agree with you. We have modified caption for Fig. 4e in the manuscript.

Comment 39: I would use the word "gas" more than "vapor" here.

Response 39: We agree “gas” is better than “vapor”, but in the cited reference, “vapor” was used. Therefore, we would keep “vapor” here.

Comment 40: I would suggest a brief comparison with the similar scenarios reconstructed for other LIPs, such as the end-Triassic CAMP (e.g., Heimdal et al., 2020; Capriolo et al., 2021).

Response 40: We have added in a comparison in the manuscript. Thank you for this suggestion.

Comment 41: Since the stability of CH₄ in the atmosphere is quite short, I would

suggest to state its severe impact on the biosphere but to deal with the atmospheric CO₂ concentration.

Response 41: This is a great point. We have added a discussion on this comment.

Comment 42: You probably mean "5.0 °C/km" instead of "5.0 °C/Ma", right?

Response 42: Here we do mean the geological heating rate of "5.0 °C/Ma" over time rather than a geothermal gradient in depth. It is used for calculating the geological generation temperatures based on a kinetic model.

Mentioned references

- Basu, A. R. et al. High-³He Plume Origin and Temporal-Spatial Evolution of the Siberian Flood Basalts. *Science* 269, 822–825 (1995).
- Ballentine, C. J., Burgess, R. & Marty, B. Tracing Fluid Origin, Transport and Interaction in the Crust. *Reviews in Mineralogy and Geochemistry* 47, 539-614 (2002).
- Gautheron, C. & Moreira, M. Helium signature of the subcontinental lithospheric mantle. *Earth and Planetary Science Letters* 199, 39-47 (2002).
- Liu, S. et al. The early Cambrian Mianyang-Changning intracratonic sag and its control on petroleum accumulation in the Sichuan Basin, China. *Geofluids* 2017, 16 (2017).
- Shellnutt, J. G. The Emeishan large igneous province: A synthesis. *Geoscience Frontiers* 5, 369-394 (2014).
- Stuart, F. M., Burnard, P. G., Taylor, R. P. & Turner, G. Resolving mantle and crustal contributions to ancient hydrothermal fluids: He-Ar isotopes in fluid inclusions from Dae Hwa W-Mo mineralisation, South Korea. *Geochimica et Cosmochimica Acta* 59, 4663-4673 (1995).
- Xu, Y., He, B., Chung, S., Menzies, M. A. & Frey, F. A. Geologic, geochemical, and geophysical consequences of plume involvement in the Emeishan flood-basalt province. *Geology* 32, 917 (2004).
- Zou, C. et al. Formation, distribution, resource potential, and discovery of Sinian-Cambrian giant gas field, Sichuan Basin, SW China. *Petroleum Exploration and Development* 41, 306-325 (2014).
- Zhu, G., Wang, T., Xie, Z., Xie, B. & Liu, K. Giant gas discovery in the Precambrian deeply buried reservoirs in the Sichuan Basin, China: Implications for gas exploration in oil cratonic basins. *Precambrian Research* 262, 45-66 (2015).

Reviewer #1 (Remarks to the Author):

I commend the authors for their comprehensive response to both my comments and the comments of the other reviewers. I am now quite happy with their conclusions and am able to follow the calculations that they did throughout. Well done.

Pete Barry, WHOI

Reviewer #2 (Remarks to the Author):

The authors have thoroughly addressed the issues raised by editors and reviewers.